# Efficacy and Safety of Habitual Consumption of a Food Supplement Containing Miraculin in Malnourished Cancer Patients: The CLINMIR Pilot Study

**DOI:** 10.3390/nu16121905

**Published:** 2024-06-17

**Authors:** Bricia López-Plaza, Ana Isabel Álvarez-Mercado, Lucía Arcos-Castellanos, Julio Plaza-Diaz, Francisco Javier Ruiz-Ojeda, Marco Brandimonte-Hernández, Jaime Feliú-Batlle, Thomas Hummel, Ángel Gil, Samara Palma-Milla

**Affiliations:** 1Food, Nutrition and Health Platform, Hospital La Paz Institute for Health Research (IdiPAZ), 28046 Madrid, Spain; lucia.arcos.castellanos@idipaz.es (L.A.-C.); samara.palma@salud.madrid.org (S.P.-M.); 2Medicine Department, Faculty of Medicine, Complutense University of Madrid, Plaza de Ramón y Cajal, s/n, 28040 Madrid, Spain; 3Department of Pharmacology, University of Granada, 18071 Granada, Spain; alvarezmercado@ugr.es; 4Instituto de Investigación Biosanitaria ibs.GRANADA, Complejo Hospitalario Universitario de Granada, 18014 Granada, Spain; jrplaza@ugr.es (J.P.-D.); fruizojeda@ugr.es (F.J.R.-O.); agil@ugr.es (Á.G.); 5Institute of Nutrition and Food Technology “José Mataix”, Centre of Biomedical Research, University of Granada, Avda. del Conocimiento s/n, Armilla, 18016 Granada, Spain; mbrandimonte@ugr.es; 6Department of Biochemistry and Molecular Biology II, University of Granada, 18071 Granada, Spain; 7Children’s Hospital of Eastern Ontario Research Institute, Ottawa, ON K1H 8L1, Canada; 8CIBEROBN (CIBER Physiopathology of Obesity and Nutrition), Instituto de Salud Carlos III, 28029 Madrid, Spain; 9Oncology Department, Hospital La Paz Institute for Health Research—IdiPAZ, Hospital Universitario La Paz, 28029 Madrid, Spain; jaime.feliu@salud.madrid.org; 10CIBERONC (CIBER Cancer), Instituto de Salud Carlos III, 28029 Madrid, Spain; 11Medicine Department, Faculty of Medicine, Autonomous University of Madrid, Arzobispo Morcillo 4, 28029 Madrid, Spain; 12Smell & Taste Clinic, Department of Otorhinolaryngology, Technische Universität Dresden, Fetscherstraße 74, 01307 Dresden, Germany; thomas.hummel@tu-dresden.de; 13Nutrition Department, Hospital University La Paz, 28046 Madrid, Spain

**Keywords:** taste disorders, ageusia, dysgeusia, neoplasm, chemotherapy, radiotherapy, *Synsepalum dulcificum*, miraculin protein, miracle berry, malnutrition, fatty acids

## Abstract

Taste disorders (TDs) are common among systemically treated cancer patients and negatively impact their nutritional status and quality of life. The novel food approved by the European Commission (EFSA), dried miracle berries (DMB), contains the natural taste-modifying protein miraculin. DMB, also available as a supplement, has emerged as a possible alternative treatment for TDs. The present study aimed to evaluate the efficacy and safety of habitual DMB consumption in malnourished cancer patients undergoing active treatment. An exploratory clinical trial was carried out in which 31 cancer patients were randomized into three arms [standard dose of DMB (150 mg DMB/tablet), high dose of DMB (300 mg DMB/tablet) or placebo (300 mg freeze-dried strawberry)] for three months. Patients consumed a DMB tablet or placebo daily before each main meal (breakfast, lunch, and dinner). Throughout the five main visits, electrochemical taste perception, nutritional status, dietary intake, quality of life and the fatty acid profile of erythrocytes were evaluated. Patients consuming a standard dose of DMB exhibited improved taste acuity over time (% change right/left side: −52.8 ± 38.5/−58.7 ± 69.2%) and salty taste perception (2.29 ± 1.25 vs. high dose: 2.17 ± 1.84 vs. placebo: 1.57 ± 1.51 points, *p* < 0.05). They also had higher energy intake (*p* = 0.075) and covered better energy expenditure (107 ± 19%). The quality of life evaluated by symptom scales improved in patients receiving the standard dose of DMB (constipation, *p* = 0.048). The levels of arachidonic (13.1 ± 1.8; 14.0 ± 2.8, 12.0 ± 2.0%; *p* = 0.004) and docosahexaenoic (4.4 ± 1.7; 4.1 ± 1.0; 3.9 ± 1.6%; *p* = 0.014) acids in erythrocytes increased over time after DMB intake. The standard dose of DMB increased fat-free mass vs. placebo (47.4 ± 9.3 vs. 44.1 ± 4.7 kg, *p* = 0.007). Importantly, habitual patients with DMB did not experience any adverse events, and metabolic parameters remained stable and within normal ranges. In conclusion, habitual consumption of a standard 150 mg dose of DMB improves electrochemical food perception, nutritional status (energy intake, fat quantity and quality, fat-free mass), and quality of life in malnourished cancer patients receiving antineoplastic treatment. Additionally, DMB consumption appears to be safe, with no changes in major biochemical parameters associated with health status. Clinical trial registered (NCT05486260).

## 1. Introduction

Taste disorders (TDs) are frequent adverse events during antineoplastic treatments in cancer patients [1,2,3,4]. However, limited attention has been given to these disorders. The effects of TDs are related to the cytotoxic effects of chemotherapy on the differentiation and proliferation of cells in the taste bud [5] or to chemosensory dysfunction that can cause neurological damage by acting directly on taste receptors or synaptic uncoupling during radiotherapy [6]. Stem cell therapy [7] and anticancer-targeted drugs [8,9] have also been shown to induce taste alterations. However, chemotherapy-related TDs are more frequent. Chemotherapy-induced TDs are highly variable and range between 17% and 86% [10]. The presence of TDs can occur as acute side effects after chemotherapy [11] increasing according to the number of cycles received. Although these symptoms generally improve once treatment is completed [12], they may also persist for a long period after treatment is completed [13]. One of the most prevalent TDs, from the qualitative and quantitative point of view, is dysgeusia, which occurs in between 56% and 76% of patients receiving antineoplastic treatment [14]. Dysgeusia is a gustatory disturbance defined as impaired or altered sense taste perception or persistent taste sensation without stimulation [15]. Generally, patients described unpleasant tastes or distortions of taste sensation [16].

Patients commonly present anorexia due to antineoplastic treatment but also due to dysgeusia. Indeed, patients attribute difficulties maintaining adequate food intake to altered taste during treatment [17]. TDs reduce appetite and energy intake, which produce changes in food preferences [18] that determine weight loss and changes in body composition [19] and increase malnutrition risk in cancer patients [20].

The prevalence of malnutrition in cancer patients varies between 40% and 80% [21]. This condition determines the outcome of cancer patients [22] since its presence is associated with treatment-induced toxicity, an increase in the postoperative risk of complications [23], poor prognosis, overall survival reduction [24], and increased mortality. In this sense, TDs can increase malnutrition risk by a factor of 3.36 [19]. TDs can also have a significant impact on cancer patients’ quality of life by reducing food enjoyment [1,11] and developing food aversions that reduce food intake [25] and increase the risk of malnutrition [26,27].

Therefore, it is not surprising that different strategies have been developed to prevent or ameliorate TDs [28,29,30,31,32]. Commonly known as the miracle berry, the *Synsepalum dulcificum* (Daniell) fruit has attracted increased attention due to its ability to transform sour taste perception into sweet taste [33]. This quality is due to the presence of miraculin, a glycoprotein that acts as a selective agonist at acidic pH or antagonist at neutral pH, of sweet taste receptors [34]. This characteristic allows miraculin to change the food flavor depending on the pH of the food consumed, making meals more palatable. Miraculin provides a high sweetness intensity that persists for approximately 30 min after consumption [35]; thus, its consumption could improve the overall taste perception in cancer patients undergoing antineoplastic treatment and those with TDs [36], improving food intake and, consequently, their nutritional and health status.

Two studies have evaluated the consumption of miracle fruit in cancer patients undergoing active chemotherapy treatment, and both have shown positive changes in TDs [37,38]. However, despite pointing out the direction of the effect of consuming the miracle berry on these patients, both studies used subjective methods for the assessment of TDs and used the fruits of *S. dulcificum*.

In December 2021, the European Commission authorized dried miracle berry (DMB) as a *novel food* [39]. DMB, is a freeze-dried extract of miracle berry pulp juice rich in miraculin. It was officially cataloged as the dried fruit of *S. dulcificum*, safe for use in the European Union. DMB has become available as a food supplement.

In this sense, the present study hypothesizes that DMB consumption enhances the electrochemical taste perception and improves both the nutritional status and quality of life of cancer patients, positively impacting their health. Therefore, the main aim of the present clinical trial was to evaluate the efficacy and safety of habitual DMB consumption in malnourished cancer patients undergoing active treatment.

## 2. Materials and Methods

A detailed description of the CLINMIR study protocol has recently been published elsewhere [40]. Below is a summary of the clinical trial.

### 2.1. Trial Design

The clinical trial protocol was approved by the Scientific Research and Ethics Committee of the Hospital University La Paz (HULP), Madrid (Spain) in version 1 in June 2022 and protocolled by the HULP Code 6164. The present protocol clinical trial has also been registered at http://clinicaltrials.gov (first posted on 3 August 2022) with the number NCT05486260.

The CLINMIR study is a pilot randomized, parallel, triple-blind, and placebo-controlled clinical trial allocated in three arms according to treatment with a food supplement enriched in the protein miraculin (DMB) in malnourished cancer patients exhibiting TDs because of active chemotherapy and radiotherapy and adjusted by type of cancer. All patients were recruited from medical consultations in the Clinical and Dietary Nutrition Unit (UNC&D) and by referral from the Oncology Service of the HULP to UNC&D.

### 2.2. Participants

The main inclusion criteria were patients 18 years of age and older with cancer, active chemotherapy and/or radiotherapy, and/or immunotherapy treatment who had a weight loss ≥ 5% in the last six months, malnutrition diagnosis assessed by Global Leadership Initiative on Malnutrition (GLIM Criteria) [41], and TDs measured by electrogustometry. Additionally, patients had to have a life expectancy greater than 3 months and be able to feed by oral intake. Patients also had an understanding of the clinical study guidelines.

The exclusion criteria included patients participating in another clinical trial, enteral or parenteral nutrition, poorly controlled diabetes mellitus (HbA1c > 8%), uncontrolled hypertension or hyper/hypothyroidism, severe digestive toxicity due to treatment with chemo-radiotherapy, severe kidney or liver disease (chronic renal failure, nephrotic syndrome, cirrhosis, etc.), severe dementia, brain metastases, eating disorders, history of severe neurological or psychiatric pathology that may interfere with treatment, alcoholism or substance abuse, severe gastrointestinal diseases, and unwillingness to consume the miraculin-based food supplement.

Intolerance to miraculin was a withdrawal criterion. Any medication that did not interfere with the study formulation was allowed and registered in the clinical research data.

### 2.3. Interventions

Patients who met the selection criteria were randomized to one of three arms of the clinical trial. The first arm had 150 mg of DMB equivalent to 2.8 mg of miraculin + 150 mg of freeze-dried strawberries per orodispersible tablet; the second arm had 300 mg of DMB equivalent to 5.6 mg of miraculin; and the third arm contained 300 mg of freeze-dried strawberries per orodispersible tablet as a placebo. All treatments were isocaloric (Table 1).

Those patients who voluntarily agreed to participate signed the informed consent form. Over 3 months, each patient consumed an orodispersible tablet containing DMB or placebo five minutes before each main meal (breakfast, lunch, and dinner).

The clinical trial had six face-to-face visits in two phases, one selection visit (vS) in the selection phase and five visits in the experimental phase (Figure 1).

On the selection visit, nutritional status was assessed according to the GLIM criteria as well as electrical (electrogustometry) and chemical taste perception (taste strips). The included patients received the questionnaires to complete and hand in at visit 1 (food daily record of 3 days, one holiday (weekend day, a day off, or a day out of the usual routine), quality of life questionnaire (EORTC QLQ-C30) and International Physical Activity Questionnaire (IPAQ) as well as the blood sample extraction appointment (analysis of biochemical parameters and fatty acids from erythrocytes). Experimental phase visits were 4–7 days after their chemotherapy infusion, except visit 1 before it.

At visit 1 (v1), patients were randomized and provided with the necessary product (DMB or placebo) until their next visit (v2). Anthropometric measurements, electrical bioimpedance, and the Sniffin’ sticks smell test were carried out. Healthy eating and physical exercise guidelines for cancer patients were explained. As part of the next visit, the following forms were delivered: a product efficacy satisfaction questionnaire, a product consumption control daily sheet, a product consumption tolerance record sheet, and a record sheet of adverse effects. Additionally, individualized nutritional treatment was implemented. If an oral nutritional supplement was needed, a polymeric, hypercaloric, and hyperproteic formula enriched in omega-3 fatty acids was prescribed depending on their energy requirements.

Visits 2 (v2, 4–7 days after the chemotherapy session), 3 (v3, ± 1 month after visit 1) and 4 (v4, ± 2 months after visit 1) were similar and they were carried out 4–7 days after the chemotherapy session. During these visits, nutritional status was monitored, and anthropometric measurements and smell and taste tests (electrogustometry, taste strips tests and Sniffin’ sticks smell test) were carried out. In these visits, biochemical parameters were also measured. Completed questionnaires were collected (food daily record, quality of life questionnaire, product efficacy, product consumption control, tolerance record and adverse effects record) and behavioral reinforcement (nutritional treatment and physical activity, consumption and registration of the assigned treatment, and tolerance and adverse effects registry). Patients received the questionnaires to complete and hand in at the next visit.

Finally, during visit 5 (v5, ± 3 months after v1 and 3–4 days after the patient’s chemotherapy), nutritional status was assessed and anthropometric measurements and taste and smell tests were carried out (electrogustometry, taste strips tests, and Sniffin’ sticks smell test). A blood sample was extracted (biochemical parameters and fatty acids from erythrocytes) for analysis. Food daily records and completed quality of life questionnaires were collected, as well as product efficacy questionnaires, product consumption control, tolerance records, and adverse effects records. Behavioral reinforcement of nutritional treatment and physical activity were carried out.

### 2.4. Outcomes

Malnourished cancer patients with TDs who consumed DMB were expected to improve their taste perception by reducing the electrical-chemical taste perception threshold from baseline (v0) and throughout the intervention. Moreover, it is expected that DMB consumption improves the chemical and olfactory perception of food. Improvements in dietary intake and nutritional and safety biochemical parameters, as well as improvements in the essential and polyunsaturated fatty acid status assessed through the fatty acid composition of erythrocytes, were expected because of a better perception of food. Tolerance and possible adverse effects were also outcomes studied since several doses were evaluated. All parameters were evaluated from baseline to the end of the intervention and evolution was measured through the different visits carried out (v1, v2, v3, v4, v5).

### 2.5. Sample Size

Because the CLINMIR study was exploratory and there was a lack of previous studies using miraculin-based nutrition supplements in cancer patients, the sample size was established by the researchers. The number established was 10 patients per arm given a sample size of 30 patients. The results obtained will be able to serve to establish the sample size needed to evaluate the efficacy of the intervention product in multicenter studies.

### 2.6. Randomization and Blinding

Randomization was carried out using computer-generated random numbers in blocks of six taking into account treatment and cancer type. This sequence was generated by the Biostatistics Unit (HULP). The allocation sequence was provided in a separate document. To implement the allocation, the sequences were sequentially numbered and sealed in envelopes that were mailed to the nutritionist who enrolled and assigned participants to interventions. When the patient signed the informed consent (v1), the patient’s randomization envelope was opened.

Researchers, trial patients, care providers (nutritionists, nurses, physicians), outcome assessors, data analysts, and the promoter were blinded after assignment to interventions. Both miraculin-based food supplements and placebo had similar appearances (pink tablets). They were packaged in white opaque bottles with 30 orodispersible tablets identified by a lot number (L01, L02, L03) and a barcode for tracking. The test product in its powder form (DMB) and the placebo were provided by Baïa Food (Medicinal Gardens SL) to Rioja Nature Pharma. The packaging, in the form of bottles equipped with protective technology for moisture and oxygen-sensitive products (Activ Vial^®^), was supplied by CSP Aptar Technologies. Rioja Nature Pharma was responsible for the manufacturing, labeling, identification, and supply of the final product, and maintained the blind throughout the study until the statistical analysis was completed.

### 2.7. Specific Methodology

#### 2.7.1. Malnutrition Criteria

Nutritional diagnosis of malnutrition was established through the GLIM criteria based on phenotypic and etiological criteria. It requires at least one phenotypic criterion and one etiologic criterion to diagnose malnutrition. Body composition by bioelectrical impedance analysis (BIA) was used to evaluate reduced muscle mass. Gastrointestinal symptoms as supportive indicators were considered to assess and evaluate reduced food assimilation and major infection. Finally, trauma or acute conditions were associated with inflammation. Malnutrition was classified as moderate or severe malnutrition [41]. Nutritional status was evaluated at all study visits.

#### 2.7.2. Anthropometric Parameters

Anthropometric parameters were taken using standard techniques, following the international norms established by the WHO. Body weight was measured using a clinical digital scale (capacity 0–150 kg). The percentage of weight loss was assessed as follows: [(current weight − weight 6 months ago)/weight 6 months ago] * 100. Height was measured with a height meter with an accuracy of 1 mm (range, 80–200 cm). Body mass index (BMI) was determined using weight (kg)/height (m)^2^. Anthropometric parameters were measured at the main visits (v1, v3, v4 and v5).

#### 2.7.3. Daily Food Record

Diet was collected in three different days’ daily food records, one of which had to be a holiday. Patients were instructed to record the weight of the food consumed or, if this was not possible, to record household measurements (spoonfuls, cups, etc.). All records were thoroughly reviewed by a nutritionist in the presence of the patient to ensure that the information collected was complete. Foods, drinks, dietary supplements, and preparations consumed were transformed into energy and nutrients using DIAL software Version 3.15 (Alce Ingeniería, Madrid, Spain). Results were compared with the recommended intakes of the Spanish population.

#### 2.7.4. Electrogustometry

The threshold for an electric-induced taste stimulus (taste acuity) was measured using an electrogustometer (SI-03 Model, Sensonics International, Haddon Heights NJ, USA). Patients were instructed not to eat or drink for an hour before electrogustometry. A monopolar electrode applied the electric stimulus. The electrogustometer produces low-amplitude stimuli of a predetermined duration (0.5 s). The methodology used was that recommended by the manufacturer. The electric threshold scores were measured in the area of the fungiform papillae on both sides of the tongue. To detect thresholds, a two-down and one-up forced-choice single staircase procedure and a stimulus-response staircase were used. Threshold differences between the left and right sides greater than 7 dB were considered abnormal [42].

#### 2.7.5. Taste Strips Test

The taste strips test is a validated method to measure chemical taste perception [30]. This tool is based on the chemical perception of taste through taste-impregnated filter paper strips (Burghart Messtechnik GmbH, Holm, Germany). Four different taste strips (sweet, sour, salty, and bitter) were measured at four different concentrations each. For the assessment of whole-mouth gustatory function, strips were placed on the tongue and savored with the closed mouth for 10 s. Once the strip was removed, the participants had to identify the taste within a forced-choice procedure. A maximum score of 16 points (four concentrations of each of the four basic taste qualities) was obtained. Hypogeusia was considered when a score below nine was obtained regardless of age.

#### 2.7.6. Sniffin’ Sticks Smell Test

Smell perception was measured based on odor-containing felt-tip pens (“Sniffin’ sticks” Burghart Messtechnik GmbH, Germany). Consuming food, drinks, or cigarettes 15 min before testing was not allowed. A total of 16 odor pens were presented to be identified. For each pen, a flash card with four choices was provided (e.g., pineapple, orange, blackberry, strawberry). Each uncovered odor pen was held 2 cm in front of the nostrils for 3–4 s. Based on the multiple forced-choice paradigm; patients had to choose the best match with their olfactory perception. The score sums all correct answers and was used to differentiate between normosmia and hyposmia depending on the age of the patient.

### 2.8. Quality of Life

This was evaluated using the EORTC QLQ-C30 questionnaire for cancer patients validated in Spanish [43]. The questionnaire is formed of five functional scales (daily activities and physical, emotional, cognitive, and social functioning), three symptomatic scales (fatigue, pain and nausea, and vomiting), one overall health scale, and six questions about dyspnea, insomnia, anorexia, constipation, diarrhea, and economic impact. All questions are about the previous week and are scored with 1 to 4 points. The last two questions have a score from 1 to 7, with 1 being terrible and 7 being excellent.

Scores obtained are standardized from 0 to 100 points to determine the disease impact on each scale. High scores on the global health status and functional scales indicate a better quality of life, while low scores on the symptoms scale indicate a decrease in quality of life.

### 2.9. Tolerance and Adverse Events

Gastrointestinal disorders such as abdominal distension, abdominal pain, nausea, regurgitation or gastroesophageal reflux, vomiting, constipation, diarrhea, and flatulence were defined and recorded based on the Common Terminology Criteria for Adverse Events (CTCAE) from the National Cancer Institute [44]. These adverse events were classified as Grade 0 (not described), Grade 1 (mild), Grade 2 (moderate), Grade 3 (severe), Grade 4 (mortality risk), and Grade 5 (death associated with an event). Additionally, the patients were asked if they could be related to product consumption.

### 2.10. Fatty Acid Profile of Erythrocytes

The separation and quantification of fatty acids from erythrocyte lipids have been reported in previous works [45]. Briefly, erythrocyte lipid extraction and fatty acid methylation were performed as described by Lepage and Roy (1988) [46]. Fatty acid methyl esters (FAME) were identified and quantified by comparing their retention times by gas chromatography-mass spectrometry (GC-MS). This analysis was performed by injecting 1 µL into a Bruker (Bremen, Germany) model 456-GC high-resolution gas chromatograph coupled to a Bruker model EVOQ TQ triple quadrupole mass spectrometer as follows:

GC conditions
(a)ZB-FAME capillary column (30 m × 0.25 mm ID × 0.20 um film).(b)Split mode injector (100:1)(c)Injector temperature: 250 °C(d)Transfer line temperature: 240 °C(e)Carrier gas: He (1 mL/min)(f)Temperature ramp: 100 °C (2 min) up to 210 °C (5 min) at 4°/min.

MS conditions:(a)Temperature of the source: 240 °C(b)Full scan from 45 Da to 450 Da(c)Electron impact ionization (EI+) at 70eVFood daily record

### 2.11. Biochemical Parameters

Biochemical analyses were carried out in the Biochemistry Laboratory of the Hospital La Paz, an ISO-certified laboratory, at each visit (v1, v3, v4, v5) using an Olympus AU5400 Automated Chemistry Analyzer (Olympus Corporation, España Barcelona, Spain).

### 2.12. Miraculin-Based Food Supplement Taste Perception

A visual analog scale (VAS) was designed by the researchers to obtain information about the miraculin-based food supplement’s taste perception efficacy. The questionnaire included five questions using 10 cm scales, where 0 means not at all or very bad and 10 means very good or very effective. The questions included were as follows: Do you notice a food taste change after consuming the product? Does food taste better to you? Does it allow you to eat more food? What is your opinion of the product? Are you satisfied with the effectiveness of the product? Does the administration of the product seem adequate to you?

### 2.13. Statistical Methods

Data analysis was carried out by the intention to treat. Quantitative data are presented as the means ± standard deviations (SD), and percentages. Data type distribution was determined using Shapiro-Wilks tests. Levene’s test was used to evaluate the homogeneity of variances. Parametric or nonparametric tests were performed depending on the data distribution. General linear mixed models (GLM) of covariance (ANCOVA) were used to evaluate the differences between means for treatment, time, and treatment x time using covariates as the baseline data. The analysis of the qualitative variables and percentages was carried out through χ2 or Fisher’s F analysis.

Double-sided tests were applied when needed, and a *p*-value < 0.05 was considered statistically significant. Data were analyzed using R Project for Statistical Computing (https://www.r-project.org/ Accessed on 1 March 2024).

## 3. Results

The recruitment period was extended from November 2022 to May 2023. A total of 62 patients were evaluated for eligibility. Of them, 31 oncologic patients met the selection criteria and were randomized into the three intervention groups, adjusted by the type of cancer (Figure 2). During follow-up, extended from November 2022 to August 2023, there were 10 dropouts, most of them due to the taste distortion of non-sweet acidic foods (n = 6) and because the prescription derived from the intervention added difficulty to their, already complex, antineoplastic treatment (n = 2). Additionally, there were two *exitus letalis* in the placebo group. There was a 32% dropout and only 21 cancer patients completed the clinical trial; however, all variables were evaluated by intention to treat analysis.

### 3.1. General Characteristics of the Population

The sample consisted of 58.1% women and 41.9% men, with a mean age of 60.0 ± 10.9 years old. All of them were undergoing active treatment with at least chemotherapy, and TDs were measured by electrogustometry (Table 2).

The average body mass index (BMI) was 22.1 ± 3.3 kg/m^2^, indicating that the patients were within the normal weight range. However, the weight loss in the last six months was −7.8 ± 6.9%, with no significant differences between treatment groups (*p* = 0.891). The most prevalent cancer type was colorectal cancer, followed by breast, lung, pancreas, and liver cancers, with no significant differences between treatments. Treatment adherence was adequate (85.6%) with no significant difference between treatments (*p* = 0.337).

### 3.2. Miraculin-Based Food Supplement Efficacy

#### 3.2.1. The Effect on Electrical and Chemical Taste Perception

Overall, the electrical taste perception did not show significant changes depending on treatment, time, and their interaction with treatment per time (Table 3, Figure 3). However, patients consuming the standard dose of DMB had the lowest detection levels at the end of the intervention and considerably reduced the taste threshold for an electric-induced taste stimulus (taste acuity) over time (% change right/left side: −52.8 ± 38.5/−58.7 ± 69.2%). None of the cancer patients reached normal thresholds once the intervention was completed (<7 dB).

However, at the end of the study, the chemical taste perception reached normal levels (≥ 9) in all patients (Table 4). When different tastes were evaluated, salty taste perception changed over time and depending on the treatment assigned (*p* < 0.001). In this regard, patients consuming DMB significantly improved the perception of salty taste versus placebo (*p* < 0.05) (Figure 4).

Particularly, those taking both DMB standard and high doses experienced a notable percentage of change from baseline (108.3 ± 134.4 and 158.3 ± 116.7%) contrasting with placebo (−22.2 ± 72.0%). Although no significant changes were observed depending on time or treatment, bitter taste, frequently affected by chemotherapy treatment, had a lower percentage change in those patients receiving the standard dose of DMB (% change = 14.3 ± 65.6%) contrasting with the high dose of DMB (25.0 ± 16.7%) or placebo (33.3 ± 94.3%). Smell perception did not change throughout the clinical trial (Table 4).

#### 3.2.2. The Effect on Dietary Intake

Since the beginning of the study, the diet of the cancer patients was high in protein and fat, and this condition persisted throughout the study. Patients consuming the standard dose of DMB declared not having consumed a smaller amount of food (*p* = 0.032) considering 22% perceived eating less at the beginning of the study (Appendix A).

Related to the above, changes in energy intake (*p* = 0.075) were observed in patients depending on treatment (Table 5). Indeed, at the end of the intervention, the group receiving the standard dose of DMB exhibited the highest energy intake compared with the other two groups. Moreover, patients consuming the standard dose of DMB were those who best-covered their energy expenditure (107 ± 19%).

The energy contribution of lipids (*p* = 0.017) and carbohydrates (*p* = 0.060) changed over time and depending on the treatment assigned. Only patients consuming the standard dose of DMB reduced the energy contribution of carbohydrates (% change = −17.6 ± 13.1). Also, these patients had a greater lipid contribution compared to those consuming the high dose of DMB (*p* = 0.003) or placebo (*p* = 0.020). In addition, patients taking the standard dose of DMB also had a greater lipid percentage change from the beginning to the end of the intervention (22.0 ± 15.7%). Moreover, there was a significant change over time and depending on treatment in the dietary percentage provided by saturated fatty acids (SFA, *p* = 0.042) and a trend in monounsaturated fatty acids (MUFA, *p* = 0.092). In this regard, patients consuming a standard dose of DMB tended to intake more SFA versus placebo (*p* = 0.071). Additionally, patients consuming the standard dose of DMB increased all major dietary fatty acids from the beginning to the end of the intervention, including SFA (% change = 11.2 ± 20.2%), MUFA (40.6 ± 33.2) and polyunsaturated fatty acids (PUFA, 41.1 ± 123.9%) different from those consuming high dose of DMB (−8.6 ± 32.7; −6.4 ± 15.4; 3.2 ± 45.6%) or placebo (−12.0 ± 16.5; 4.1 ± 20.6; 1.9 ± 29.9%).

Taking the latter into account, after three months of intervention, all patients showed a trend to decrease in levels of palmitic and stearic acid from the fatty acid profile of erythrocytes (*p* = 0.068), particularly in DMB patients (% change standard dose: −13.2 ± 44.0/−7.9 ± 17.5; high dose: −7.9 ± 29.1/−11.9 ± 19.7; placebo: 3.6 ± 35.9/−1.0 ± 42.3) (Table 6, Figure 5). In patients consuming standard and high doses of DMB, the increase in linoleic acid percentage change was 15.3 ± 15.0 and 4.7 ± 17.3%, respectively, while it was reduced in placebo (−6.0 ± 20.7%).

Moreover, there was a change in total PUFA (*p* = 0.009), total PUFA *n*-6 (*p* = 0.010), arachidonic acid (20:4 *n*-6, *p* = 0.004), EPA (20:5 *n*-3, *p* = 0.093), DHA (22:6 *n*-3, *p* = 0.014) and omega-3 index (*p* = 0.010) over time. In the groups consuming standard and high doses of DMB, arachidonic acid (AA) increased the percentage change by 49.9 ± 57.9 and 42.1 ± 49.8%, respectively, while in placebo by 8.4 ± 31.5%. The percentage of change of *n*-6 PUFA was higher in patients consuming the standard dose of DMB (30.2 ± 26.0%) and high doses of DMB (23.5 ± 28.5%) in contrast to placebo (1.5 ± 26.6%). It was also the standard dose of DMB consumed by cancer patients who observed a greater percentage of change in DHA (81.2 ± 94.7%) and omega-3 index (52.7 ± 81.0%) from the beginning to the end of the intervention.

#### 3.2.3. The Effect on Anthropometry and Body Composition

After three months of intervention with the miraculin-based food supplement, oncologic patients tended to change body weight (*p* = 0.073), BMI (*p* = 0.073), and waist circumference (*p* = 0.053) and significantly changed fat-free mass (*p* = 0.006) and total water (*p* = 0.029) over time and depending on treatment (Table 7). Patients consuming the standard dose of DMB were those who had a higher percentage of change in body weight (−1.9 ± 4.4%), BMI (−1.4 ± 4.6%), and waist circumference (−2.4 ± 6.7%) compared to the beginning of the intervention. Compared to placebo, patients consuming the standard dose of DMB increased fat-free mass (*p* = 0.007) and those with the high dose of DMB had greater total water (*p* = 0.020). Only patients consuming DMB reduced fat mass, mainly those with a standard dose [(−2.5 ± 1.3 vs. −1.3 ± 3.2 vs. 0.5 ± 0.8 kg); (% change = −11.4 ± 35.0 vs. −6.1 ± 19.5 vs. 2.1 ± 14.8%)] (Table 7).

When the bioimpedance phase angle was evaluated, all patients showed a loss of cellular integrity throughout the study (< 5°). However, when the angle phase was standardized by age and sex, patients treated with DMB tended to present an improvement depending on treatment (*p* = 0.072). Also, the percentage of change was greater in patients consuming the standard dose of DMB (61.8 ± 19.1%) than in those with a high dose of DMB (53.7 ± 99.6%) or placebo, where it worsened (−20.6 ± 95.6%) (Table 7).

After three months of intervention, all patients regained part of the weight lost during the last 6 months before the start of the study and improved their nutritional status without significant differences between treatments (Appendix A). Two patients consuming a standard dose of DMB continued with severe malnutrition after the study ended.

#### 3.2.4. The Effect on Quality of Life

Although the global health status perception was not modified by the consumption of the miraculin-based food supplement, changes were observed on social (*p* = 0.018), fatigue (*p* = 0.044), and constipation (*p* = 0.048) scales depending on treatment (Table 8). At the end of the intervention, patients consuming a high dose of DMB significantly reduced their social scale (*p* < 0.05) and felt more fatigue (*p* < 0.05) compared to a standard dose of DMB and placebo. Patients consuming a standard dose of DMB significantly improved the presence of constipation compared to the other two groups (*p* < 0.05). In this regard, cancer patients consuming the standard dose of DMB showed a higher percentage change in the social functional scale (19.6 ± 40.3%) and constipation (−66.7 ± 57.7%) from the beginning to the end of the intervention.

Over time, a trend of change was also observed on the physical (*p* = 0.083), emotional (*p* = 0.074), and loss of appetite (*p* = 0.070) scales (Table 8). Consistent with food consumed perception, those patients consuming the standard dose of DMB showed a higher fall in loss of appetite (% change = −100.0 ± 0.0) contrasting with those consuming the high dose (−33.3 ± 57.7) or placebo (−33.3 ± 57.7%) that increased their inappetence. These patients also were the only ones who improved their emotional scale during the intervention (% change = 1.2 ± 9.9%).

In addition to a better quality of life, the perception perceived by cancer patients about product effectiveness tended to improve depending on treatment (*p* = 0.074) (Appendix A). Patients consuming the standard dose of DMB showed a better perception of its effectiveness from the start to end of intervention (% change = 44.2 ± 73.5 vs. 14.6 ± 75.3 or placebo −21.4 ± 44.4%).

### 3.3. Miraculin-Based Food Supplement Safety

#### 3.3.1. Adverse Events

During the study, some adverse events occurred in the patients evaluated (Appendix A). However, when patients were asked about the possible association with DMB consumption, all declared none of them were associated with these adverse events. Indeed, the *intensity* of adverse events reported by cancer patients consuming DMB improved once the intervention was completed. In this sense, patients who initially reported a moderate intensity changed from having a moderate intensity to mild or *not described*. Symptoms such as abdominal distention improved only in those patients consuming the standard dose of DMB. When an adverse event occurred, oncologic patients consumed the medication indicated by the physician. Thus, after three months of treatment, patients consuming DMB did not present more adverse events than those consuming placebo.

#### 3.3.2. Biochemical Parameters

Glucose metabolism parameters remained within normal ranges in all considered groups (Table 9). It is worth mentioning that, in patients consuming the standard dose of DMB, the percentage of change in insulin concentration since the beginning of the intervention was −20.8 ± 39.7%, while in the high-dose group it was −1.6 ± 50.2%, and in placebo −7.5 ± 23.4%.

Even though the diet of patients consuming the standard dose of DMB was high in fat (Table 5), the blood lipid profile was not altered, and parameters related to lipid metabolism remained within normal ranges for the age and sex of the population (Table 9).

Proteins usually related to nutritional status, such as retinol-binding protein (RBP), showed changes over time and depending on treatment (*p* = 0.027). Patients consuming the high dose of DMB had higher RBP values than placebo (*p* < 0.05); however, the mean of this increase remained within normal ranges.

Vitamin and mineral biomarkers, except for magnesium, were not affected by habitual consumption of the miraculin-based food supplement and remained stable throughout the clinical trial and within the normal ranges of the population throughout the clinical trial (Appendix A). Magnesium showed a change throughout the study depending on the time and treatment assigned (*p* = 0.028). Only those patients consuming DMB improved magnesium concentration at the end of the study (% change standard dose 4.2 ± 5.7; high dose: 11.7 ± 13.6; placebo −3.0 ± 12.7).

At the end of the study, kidney function biomarkers such as creatinine (*p* = 0.054), glomerular filtration rate (*p* = 0.051), and uric acid (*p* = 0.066) tended to change over time and depending on treatment (Table 10). Nevertheless, all patients had values within normal ranges.

Finally, safety biomarkers of liver function did not show significant changes after completing the clinical trial (Table 10), except for ALT levels (*p* = 0.057). Only patients consuming the standard dose of DMB reduced ALT levels from the beginning to the end of the intervention (% change = −7.5 ± 23.4%, high dose: 16.7 ± 32.9%, and placebo: 5.6 ± 23.2%) within the normal range while this was not so in patients consuming the placebo who had final ALT blood concentrations higher than normal (< 35 UI/L). From the beginning of the intervention to the end, lactate dehydrogenase (LDH), an enzyme used to detect tissue or liver damage, had higher levels than those recommended (100–190 UI/L) in all patients.

It is worth mentioning that, although there were no differences depending on time or treatment, at the end of the intervention the gamma-glutamyl transferase (GGT), a biomarker of possible damage to the bile ducts, was normal only in those patients consuming the standard dose of DMB, while the rest were above normal ranges (>38 IU/L).

## 4. Discussion

The main findings of the present study were that the habitual intake of a standard dose of DMB improved the electrochemical perception of taste in cancer patients allowing a greater food intake and a better quantity and quality of dietary lipid intake, which in turn was reflected in an ameliorated fatty acids status. Additionally, improvements in body composition, nutritional status, and quality of life were observed. Furthermore, the main safety parameters remained stable and within normal ranges throughout the entire study. These results suggest that the habitual consumption of a standard dose of 150 mg of miraculin food supplement (DMB) is effective and safe for malnourished cancer patients in active treatment who present with objective TDs.

Two clinical trials have been carried out on patients receiving chemotherapy using the miracle berry. In the first study, a crossover clinical trial was carried out on 23 chemotherapy patients whose taste alterations were measured by the Wickham questionnaire [37]. In two weeks, patients consumed either the miracle fruit or supportive measures alone. At the end of the study, 30% of patients showed an improvement in taste. The second study included eight participants who received three or more cycles of chemotherapy and expressed positive taste changes to the nurse [38]. These patients were assigned to the experimental (n = 4) or control group (n = 4) in a nonrandomized manner. Patients consumed six fruits per day of miracle fruit or dried cranberries as a placebo for two weeks. At the end of the study, all patients reported positive taste changes with miracle fruit consumption through qualitative data.

In the present study, a reduction in the electrical threshold (taste acuity) was observed in all patients evaluated, including those consuming the miraculin-based food supplement. This finding is relevant because a gradual deterioration in taste perception is expected to occur because of antineoplastic treatment [3,4,47,48] and this deterioration has remained stable throughout the study. Although the overall change in electrical taste perception change was not conclusive, the chemical perception of salty taste significantly improved in cancer patients habitually consuming the standard dose of DMB. Analysis of subjective taste changes reported that salt and umami tastes are more sensitive to chemotherapy than other taste descriptors [49]. Salty taste distortion is the most frequently reported taste alteration during neo/adjuvant chemotherapy [50]. Umami taste was not evaluated as a descriptor in the present clinical trial because foods providing umami flavor are not commonly used in the Spanish population. Since one taste perception is associated with changes in other tastes during chemotherapy [51] an improvement in an affected descriptor can contribute to a better perception of global food taste.

Up to 87% of cancer patients with TDs experience a loss of appetite [52] which is widely known to be associated with poor prognosis [53]. However, patients who consumed the standard dose of DMB did not exhibit a loss of appetite at the end of the study. Therefore, habitual consumption of a standard dose of DMB may protect against loss of appetite in cancer patients; in fact, these patients had greater food intake and better met their energy needs. This finding is of relevance since cancer patients have shown a lower intake of total energy, protein and fat during chemotherapy related to TDs [54].

In addition to better covering the total energy expenditure, habitual consumption of a standard dose of DMB was associated with increased quantity and quality of fat intake in cancer patients. Various studies have shown that high-fat diets, especially those rich in trans and saturated fat, promote tumorigenesis by modulating the gut microbiota [55,56,57], systemic low-grade inflammation [58], and changes in the adipocytokine profile [59,60]. On the other hand, although epidemiological data do not support the theory that a decrease in total fat intake is effective in preventing cancer [61,62,63,64] or decreasing cancer-specific mortality [65], dietary lipid composition can have an impact on cancer pathogenesis [66]. Thus, cancer patients who consumed a standard dose of DMB exhibited notably improved MUFA and PUFA intake. MUFA intake has been inversely associated with decreased cancer risk [61,67]. Indeed, a higher intake of MUFA from plant sources was associated with lower mortality rates associated with all causes [68]. Olive oil is the largest contributor to MUFA since it provides up to 78% of oleic acid, the most abundant MUFA in the Spanish diet [69]. Thus, olive oil was the most commonly used culinary fat by cancer patients in the present study. A meta-analysis of case-control studies showed that olive oil consumption was associated with lower odds of developing any type of cancer [70], which highlights the importance of its consumption.

On the other hand, a majority of studies examining the relationship between PUFAs and cancer risk have focused on *n*-6 and *n*-3, two of their most biologically active representatives. However, a meta-analysis of observational studies revealed a mild inverse association between diets high in total PUFA and specific-cancer risk [71], while others have not found an association with increased risk [64,72]. Therefore, an adequate quantity and quality of dietary fats, promoted by the habitual consumption of a standard dose of DMB, could improve the prognosis of these patients.

As shown in the present study, erythrocyte percentages of oleic acid and selected PUFA, including linoleic acid, AA, and DHA, increased following habitual intake of a standard dose of DMB. Additionally, cancer patients who consumed DMB had the highest omega-3 index, an indicator of omega-3 status and coronary heart disease risk [73]. A higher omega-3 index has also been found to be inversely associated with lower cancer-specific risk in a meta-analysis of case-control studies [74]. PUFA play important roles as precursors of lipid mediators that regulate metabolic pathways and inflammatory responses, oxidative stress, and modifications of membrane composition that could impact cell signaling pathways and cancer progression [75]. In addition, cancer cells with more membranes are less susceptible to oxidative stress induced by chemotherapeutic agents [76].

On the other hand, cancer patients undergoing chemotherapy often suffer from nutritional alterations, particularly in terms of essential fatty acid and long-chain PUFA status [77]. Additionally, nutritional status is associated with poor prognosis, lower treatment completion and greater healthcare consumption [78]. Accordingly, it has been reported that supplementation with EPA and DHA in cancer patients has a positive impact during treatment, which is associated with cellular membrane modulation [79]. Moreover, the discovery of pro-resolution mediators of inflammation derived from arachidonic acid, called lipoxins, and from EPA and DHA, called resolvins, protectins and maresins [80,81,82], supports the idea that a PUFA-enriched membrane could be favorable for the management of this disease [83,84]. In this scenario, it is possible to assume that consuming more and better-quality food would involve the intake of more essential fatty acids and lead to an improvement in the levels of PUFA with a concomitant improvement in nutritional status [85]. Changes in the fatty acid profile of the erythrocyte membrane would be indicative of improved nutritional status in cancer patients. This improvement can be attributed to supplementation with the miraculin food supplement given that it was extended for 12 weeks, sufficient time for the complete renovation of the total pool of erythrocytes [86].

In a randomized clinical trial carried out on malnourished cancer patients, a high-fat diet provided improved weight control, fat-free mass and body mass for eight weeks from the first to the third chemotherapy cycle [87]. In this regard, in the present clinical trial, habitual consumption of a standard dose of DMB maintained body weight and increased fat-free mass, as measured by BIA, a reliable tool in nutritional intervention studies [88]. This is probably because a high-fat diet, favored by the consumption of DMB, would compensate at least in part for the rise in resting energy expenditure observed in cancer patients [64], which is also a major determinant of the development of malnutrition [89]. Calorie intake is also a significant factor in preventing fat-free mass weight loss in cancer patients [90], and those consuming a standard dose of DMB adequately meet their energy requirements.

Malnutrition predicts the risk of physical impairment, chemotherapy toxicity and mortality in cancer patients [91,92]. In this sense, all cancer patients improved their nutritional status once the intervention was completed. Loss of body weight (skeletal muscle and body fat) is associated with a reduction in quality of life [93]. The latter is also affected by the disease itself and the antineoplastic treatment used [94]. Therefore, it is not surprising that poor quality of life in cancer patients is associated with poor nutritional status [95] and conversely, that malnutrition reduces their quality of life [96]. Additionally, quality of life can significantly impact long-term cancer survivorship [97]. In this regard, in the present clinical trial, it was found that habitual consumption of a standard dose of DMB improved quality of life, in particular constipation, as measured by symptom scales. Diverse catabolic factors are activated by the presence of constipation, fatigue, nausea, vomiting and other relevant symptoms usually present in cancer patients [98]. Fatigue or loss of appetite are among the most common symptoms exhibited by cancer patients that affect their quality of life [99]. In the present study, only patients who consumed a standard dose of DMB improved their loss of appetite and improved their scores on the emotional scale from the beginning to the end of the intervention. They also showed improvements in fatigue. Since TDs caused by cancer therapies negatively affect patient quality of life [14,26,52], the improvement observed in the perception of salty taste in patients consuming a standard dose of DMB could have contributed to the improvement of these quality-of-life scales.

*Synsepalum dulcificum* fruits have been consumed since the 18th century by natives of Western and Central Africa [100] without describing adverse events beyond wanted taste changes. In 2021, DMB obtained from dried fruits of *S. dulcificum* was approved as a *novel food* in the European Union after a positive scientific opinion by the European Food Safety Authority (EFSA). The panel concluded that an intake of 10 mg/kg body weight (bw) per day is safe for human consumption [39]. The maximum dose used in the present clinical trial was 0.9 g/day, slightly above this recommendation. However, the EFSA also indicated that a 90-day oral dose of 2000 mg/kg bw per day was not associated with adverse effects. In this vein, different studies assessed the taste-modifying properties of different products from *S. dulcificum* and although this has not been its main objective, the authors of these studies did not report adverse events during its consumption [38,101,102,103,104,105]. The potential allergenicity and toxicity of miraculin have also been evaluated and it has not been associated with any safety concerns [106].

In this regard, cancer patients who habitually consumed DMB did not experience any adverse events related to their consumption. A negative effect, but not an adverse event, was the dropout of six patients due to the taste distortion caused by habitually non-sweet acidic foods such as tomatoes and salads. The majority of dropouts (67%) occurred at a high dose of DMB, indicating that patients are more likely to accept a standard dose of DMB. Indeed, the effectiveness perceived by patients of the food supplement containing miraculin increased notably in those patients consuming a standard dose of DMB over time. Several studies have shown that the degree of the taste-modifying effect of miracle berries differs according to fruit type, source, or preparation [107], since it determines the miraculin content. The smaller the quantity, the lower the sweetness intensity, and vice versa [105]. A high dose of DMB, with a higher miraculin content, probably provided high sweetness intensity and persistence, significantly modifying the cancer patient’s taste of sour foods. This is because miraculin stimulates a sweet taste 400,000 times greater than sucrose [108] and its effect can linger up to two hours until miraculin dissociates from the taste receptors by the action of salivary amylase [109].

While the energetic contribution of dietary lipids increased significantly in those consuming the standard dose of DMB, its continued consumption for 3 months did not alter the blood lipid profile. Triterpenoids isolated from the miracle fruit can act as cholesterol-lowering agents [110] and as effective antihyperglycemic agents [111] by increasing insulin synthesis, inhibiting carbohydrate metabolizing enzymes [112] and improving insulin sensitivity [113]. In this regard, it is worth mentioning that the plasma lipid profile, as well as glucose metabolism parameters, remained stable and within normal ranges throughout the intervention.

Habitual consumption of a standard dose of DMB may have a hepatoprotective effect since the placebo patients had liver markers such as ALT and GGT above normal ranges. The hepatoprotective effect of miracle berries has already been described in previous experimental studies [111]. Kidney protection was also observed when miracle fruit extracts were used. Indeed, it has been proposed as a novel plasma uric-lowering agent [114]. In this sense, it was observed that patients consuming a standard dose of DMB tended to reduce the concentration of uric acid within normal ranges.

The major strength of the present clinical trial was the use of objective analysis in the evaluation of the effect of habitual consumption of a food supplement containing miraculin on electrochemical taste perception in cancer patients undergoing active treatment. Due to the exploratory nature of the present study, one of the limitations was the reduced number of patients evaluated. Additionally, the complexity of managing cancer patients (polypharmacy, complications, intercurrent diseases, etc.) may have conditioned the high treatment dropout rate. However, based on the results obtained in the present study, the calculation of the ideal sample size will allow us to confirm and expand the results in future clinical trials as the optimal dose of DMB has now been established.

## 5. Conclusions

The habitual consumption of a standard dose of DMB, equivalent to 150 mg of the miracle dried berries, before each main meal, improves electrochemical food perception allowing for greater food intake and a better quantity and quality of the lipid profile reflected in the diet and membrane fatty acids. Additionally, a standard dose of DMB increases fat-free mass and reduces fat mass but also promotes improvements in quality of life, such as constipation. The nutritional status of cancer patients who consumed a standard dose of DMB also improved. Additionally, the habitual consumption of DMB appears to be safe with no changes in major biochemical parameters associated with health status.

## Figures and Tables

**Figure 1 nutrients-16-01905-f001:**
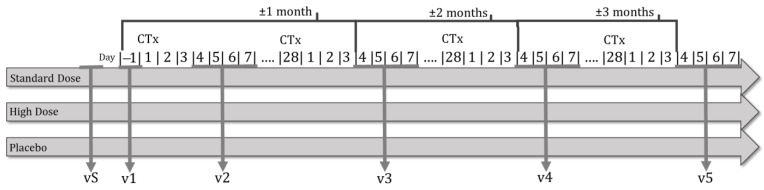
CLINMIR clinical trial outline.

**Figure 2 nutrients-16-01905-f002:**
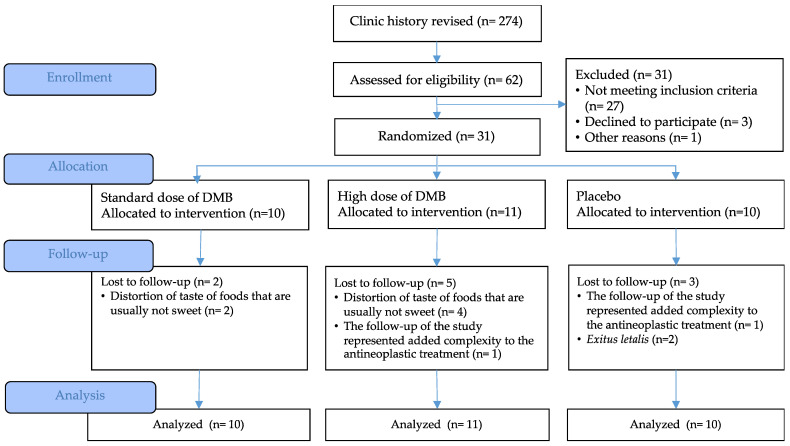
CONSORT flow diagram.

**Figure 3 nutrients-16-01905-f003:**
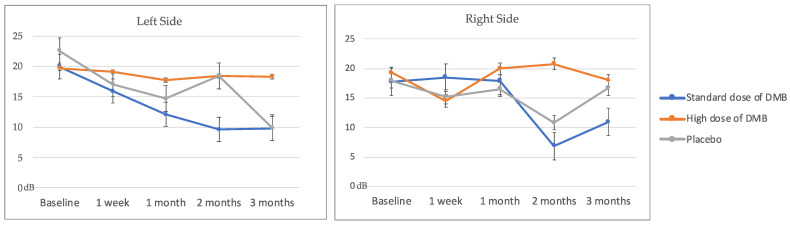
Left and right electrical taste perception (mean ± standard error).

**Figure 4 nutrients-16-01905-f004:**
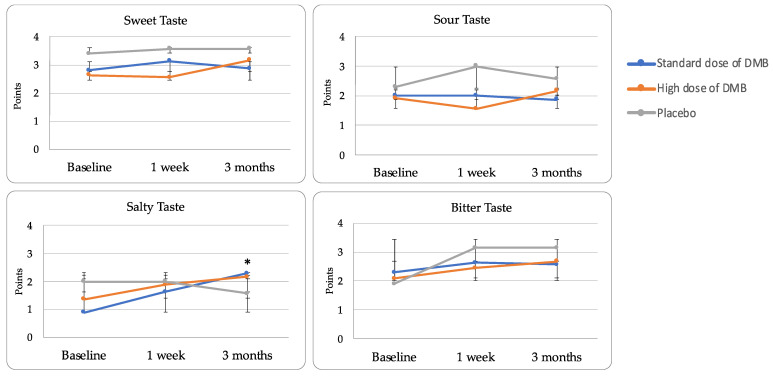
Chemical taste perception (mean ± standard error).

**Figure 5 nutrients-16-01905-f005:**
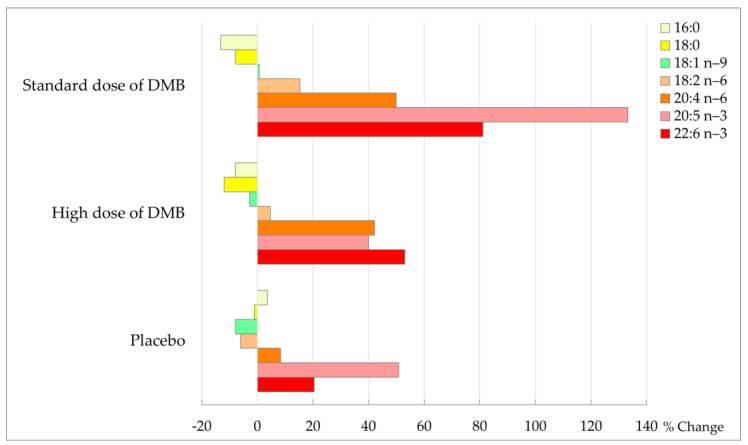
Percentage of change in membrane fatty acids at the end of the intervention (%).

**Table 1 nutrients-16-01905-t001:** Nutritional composition of the food supplement enriched in miraculin (DMB) and placebo.

		Standard Dose of DMB (150 mg DMB + 150 mg Strawberry Freeze-Dried)	High Dose of DMB (300 mg DMB)	Placebo (300 mg Strawberry Freeze-Dried)
Energy	kcal	0.99	1	0.97
Carbohydrates	mg	194	234	154
Sugars	mg	156	162	150
Fiber	mg	26	6	46
Proteins	mg	20	15	24
Lipids	mg	9	5	12
Saturated fatty acids	mg	2	2	1
Sodium chloride	mg	0.1	0.1	0.03
Humidity	mg	4	4	5
Ash	mg	12	14	15
Miraculin	mg	2,8	5,6	0

Nutritional composition provided by Medicinal Gardens, S.L.

**Table 2 nutrients-16-01905-t002:** Baseline characteristics of the population.

		Standard Dose of DMB	High Dose of DMB	Placebo	*p*-Value
Sex	Female (%)	70	45.5	60	0.517
	Male (%)	30	54.5	40	
Age	years	59.9 ± 15.1	58.9 ± 4.9	61.3 ± 11.2	0.891
Weight	kg	61.4 ± 11.1	62.0 ± 14.1	62.6 ± 10.7	0.941
Weight lost in last 6 mo.	%	7.5 ± 6.0	8.7 ± 7.1	7.2 ± 8.0	0.868
BMI	kg/m^2^	21.9 ± 3.6	22.0 ± 3.3	22.9 ± 3.4	0.737
Type of cancer					
Head and neck	%	0	9.1	0	0.895
Colorectal	%	30	27.3	20	
Esophagus	%	10	0	10	
Stomach	%	0	9.1	10	
Liver	%	0	9.1	10	
Breast	%	10	18.2	10	
Neuroendocrine	%	10	0	0	
Ovary	%	10	18.2	0	
Pancreas	%	10	9.1	10	
Lung	%	10	0	10	
Others	%	10	0	20	
Chemotherapy	%	100	100	100	1
Radiotherapy	%	20	12.5	0	0.594

BMI, body mass index. Values are expressed as mean ± standard deviation.

**Table 3 nutrients-16-01905-t003:** Electrical taste perception depending on treatment.

						*p*-Value		
			Standard Dose of DMB	High Dose of DMB	Placebo	Time (t)	Treatment (T)	T × t
Right side	(dB)	Baseline	17.7 ± 13.2	19.3 ± 14.0	17.9 ± 13.4	0.200	0.393	0.499
		1 week	18.5 ± 10.4	14.5 ± 15.5	15.2 ± 13.5			
		1 month	17.9 ± 16.3	20.0 ± 15.4	16.5 ± 17.3			
		2 months	6.9 ± 10.8	20.8 ± 14.1	10.8 ± 11.9			
		3 months	10.9 ± 11.1	18.0 ± 18.8	16.7 ± 17.1			
Left side	(dB)	Baseline	20.0 ± 12.5	19.7 ± 14.0	22.6 ± 13.8	0.444	0.544	0.946
		1 week	15.9 ± 12.9	19.1 ± 16.0	17.1 ± 15.8			
		1 month	12.1 ± 15.3	17.7 ± 15.1	14.7 ± 15.4			
		2 months	9.6 ± 13.5	18.4 ± 16.2	18.4 ± 13.1			
		3 months	9.8 ± 13.5	18.3 ± 18.4	9.9 ± 12.5			

Values are expressed as mean ± standard deviation.

**Table 4 nutrients-16-01905-t004:** Chemical perception (taste strips test) and olfactory perception (smell) depending on the treatment.

		Standard Dose of DMB			High Dose of DMB			Placebo			*p*-Value		
		Baseline	1 Week	3 Months	Baseline	1 Week	3 Months	Baseline	1 Week	3 Months	Time (t)	Treatment (T)	T × t
Chemical Taste Perception	points	8.00 ± 3.53	9.38 ± 4.24	9.63 ± 3.93	8.00 ± 3.9	8.56 ± 4.83	10.17 ± 4.67	9.6 ± 4.35	11.13 ± 3.23	10.71 ± 3.09	0.444	0.133	0.663
Sweet	right	2.8 ± 1.48	3.13 ± 1.64	2.86 ± 1.46	2.64 ± 1.5	2.56 ± 1.74	3.17 ± 1.6	3.4 ± 0.84	3.57 ± 0.79	3.57 ± 0.79	0.405	0.534	0.821
Sour	right	2.0 ± 0.82	2.00 ± 0.93	1.86 ± 0.9	1.91 ± 1.14	1.56 ± 1.24	2.17 ± 0.98	2.3 ± 1.06	3.00 ± 0.82	2.57 ± 0.98	0.194	0.688	0.591
Salt	right	0.9 ± 0.99	1.63 ± 1.41	2.29 ± 1.25	1.36 ± 1.21	1.89 ± 1.45	2.17 ± 1.84	2.00 ± 1.41	2.00 ± 1.00	1.57 ± 1.51	0.714	0.001	0.001
Bitter	right	2.3 ± 0.95	2.63 ± 1.30	2.57 ± 1.4	2.09 ± 1.22	2.44 ± 1.42	2.67 ± 1.51	1.9 ± 1.85	3.14 ± 1.22	3.14 ± 0.69	0.782	0.964	0.278
Smell Perception	points	13.2 ± 1.9	12.1 ± 2.2	12.3 ± 2.7	13.6 ± 2.4	13.8 ± 1.6	13.3 ± 2.4	12.5 ± 2.2	12.1 ± 2.4	13.0 ± 2.1	0.166	0.930	0.142

Values are expressed as mean ± standard deviation.

**Table 5 nutrients-16-01905-t005:** Diet characteristics depending on the assigned treatment.

		Standard Dose of DMB				High Dose of DMB				Placebo				*p*-Value		
		Baseline	1 Month	2 Months	3 Months	Baseline	1 Month	2 Months	3 Months	Baseline	1 Month	2 Months	3 Months	Time (t)	Treatment (T)	T × t
Intake	kcal/d	2512± 569	2641 ± 384	2364 ± 594	2679 ± 625	2254 ± 663	2030 ± 577	1809 ± 291	1850 ± 778	2338 ± 724	2294 ± 751	2035 ± 301	2443 ± 581	0.290	0.075	0.907
Contribution	%	100 ± 22	101 ± 26	90 ± 25	107 ± 19	83 ± 20	69 ± 22	62 ± 12	61 ± 29	89 ± 29	90 ± 30	79 ± 13	93 ± 26	0.513	0.324	0.982
Calorie profile																
Proteins	%	15.8 ± 2.2	16 ± 3.2	16.7 ± 2.0	16.1 ± 1.5	16.5 ± 1.72	19.4 ± 5.3	18.3 ± 2.2	18.1 ± 3.4	18.6 ± 3.5	17.4 ± 2.2	18.2 ± 2.0	15.6 ± 2.6	0.113	0.332	0.164
Carbohydrates	%	37.2 ± 4.8	34.6 ± 2.9	34.2 ± 7.1	30.4 ± 4.5	37.7 ± 7.88	36.0 ± 11.1	40.4 ± 9.0	39.3 ± 4.8	36.0 ± 5.4	40.2 ± 3.4	36.2 ± 6.1	37.5 ± 6.3	0.208	0.806	0.060
Lipids	%	41.7 ± 4.5	44.0 ± 5.3	45.3 ± 4.1	48.37 ± 5.0	42.8 ± 6.81	42.0 ± 8.	37.8 ± 10.2	39.5 ± 5.3	43.3 ± 6.0	39.4 ± 4.4	43.1 ± 6.7	43.4 ± 3.4	0.163	0.431	0.017
Lipidic profile																
SFA	%	13.8 ± 2.0	12.7 ± 2.4	13.4 ± 2.5	14.9 ± 3.3	12.8 ± 2.7	14.3 ± 4.7	11.8 ± 3.9	12.3 ± 4.0	13.5 ± 2.2	11.8 ± 3.5	13.7 ± 3.7	12.0 ± 3.1	0.483	0.799	0.042
MUFA	%	18.2 ± 4.3	20.9 ± 3.5	20.4 ± 4.2	23.1 ± 4.4	18.9 ± 4.3	17.5 ± 4.9	17.2 ± 6.8	16.3 ± 4.3	19.4 ± 5.2	18.5 ± 3.8	19.6 ± 3.8	21.0 ± 3.7	0.308	0.401	0.092
PUFA	%	7.8 ± 9.6	41.8 ± 109.1	31.1 ± 60.1	22.9 ± 46.3	29.9 ± 68.9	46.8 ± 111.8	53.7 ± 120.9	29.9 ± 59.1	18.8 ± 38.7	13.4 ± 24.8	14.1 ± 23.7	29.0 ± 57.5	0.849	0.587	0.590

SFA, saturated fatty acids; MUFA, monounsaturated fatty acids; PUFA, polyunsaturated fatty acids. Values are expressed as mean ± standard deviation.

**Table 6 nutrients-16-01905-t006:** Fatty acid profile of erythrocytes depending on treatment.

		Standard Dose of DMB		High Dose of DMB		Placebo		*p*-Value		
		Baseline	3 Months	Baseline	3 Months	Baseline	3 Months	Time (t)	Treatment (T)	T × t
Palmitic acid (16:0)	%	26.4 ± 1.8	22.4 ± 11.3	29.7 ± 8.3	26.5 ± 2.6	26.3 ± 2.7	26.1 ± 7.5	0.342	0.483	0.814
Stearic acid (18:0)	%	20.4 ± 1.4	18.2 ± 2.7	21.8 ± 4.6	18.6 ± 2.5	21.1 ± 3.0	19.8 ± 4.3	0.068	0.676	0.817
Oleic acid (18:1 *n*-9)	%	19.7 ± 3.1	20.0 ± 2.1	18.7 ± 3.2	17.8 ± 2.8	18.2 ± 1.4	17.1 ± 1.2	0.497	0.122	0.787
Total PUFA	%	41.2 ± 1.4	47.1 ± 5.2	40.7 ± 4.6	45.2 ± 5.9	41.3 ± 2.8	41.9 ± 3.1	0.009	0.759	0.784
Linoleic acid (18:2 *n*-6)	%	8.0 ± 0.8	9.0 ± 0.8	8.2 ± 0.6	8.6 ± 1.7	8.8 ± 0.9	8.4 ± 2.0	0.327	0.949	0.296
Arachidonic acid (20:4 *n*-6)	%	10.0 ± 2.8	13.1 ± 1.8	10.3 ± 2.1	14.0 ± 2.8	11.1 ± 2.0	12.0 ± 2.0	0.004	0.810	0.396
Eicosapentaenoic acid (20:5 *n*-3)	%	0.4 ± 0.2	0.6 ± 0.4	0.5 ± 0.1	0.7 ± 0.4	0.3 ± 0.2	0.5 ± 0.3	0.093	0.536	0.963
Docosahexaenoic acid (22:6 *n*-3)	%	3.0 ± 1.3	4.4 ± 1.7	3.0 ± 1.1	4.1 ± 1.0	2.9 ± 0.7	3.9 ± 1.6	0.014	0.923	0.836
Omega-3 Index	%	3.4 ± 1.3	5.0 ± 2.1	3.5 ± 1.2	4.8 ± 1.2	3.2 ± 0.9	4.4 ± 1.4	0.010	0.936	0.947

Values are expressed as mean ± standard deviation.

**Table 7 nutrients-16-01905-t007:** Anthropometric and body composition parameters depending on the assigned treatment.

		Standard Dose of DMB				High Dose of DMB				Placebo				*p*-Value		
		Baseline	1 Month	2 Months	3 Months	Baseline	1 Month	2 Months	3 Months	Baseline	1 Month	2 Months	3 Months	Time (t)	Treatment (T)	T × t
Peso	kg	61.4 ± 11.1	60.9 ± 11.6	59.8 ± 12.5	59.6 ± 12.6	62.0 ± 14.1	65.4 ± 14.3	67.5 ± 13.3	65.5 ± 14.2	62.6 ± 10.7	60.1 ± 11.8	60.8 ± 11.6	61.5 ± 11.2	0.450	0.516	0.073
BMI	kg/m^2^	21.9 ± 3.6	21.4 ± 3.1	21.0 ± 3.4	20.9 ± 3.6	22.0 ± 3.3	22.7 ± 3.3	23.0 ± 3.2	22.7 ± 3.4	22.9 ± 3.4	23.3 ± 3.4	23.6 ± 3.2	23.9 ± 3.0	0.56	0.266	0.073
WC	cm	80.5 ± 9.7	81.6 ± 11.6	78.8 ± 12.6	77.8 ± 11.9	84.3 ± 12.8	87.9 ± 12.6	90.2 ± 13.6	89.4 ± 11.4	79.4 ± 10.5	83.1 ± 10.4	83.7 ± 10.2	86.0 ± 11.6	0.857	0.209	0.053
FFM	kg	46.8 ± 8.3	48.7 ± 8.7	48.4 ± 9.4	47.4 ± 9.3	47.6 ± 9.9	48.4 ± 7.4	50.9 ± 8.5	47.3 ± 7.1	45.7 ± 5.9	43.4 ± 5.0	43.3 ± 5.2	44.1 ± 4.7	0.017	0.346	0.006
FM	kg	14.6 ± 5.5	12.1 ± 3.9	11.4 ± 4.5	12.1 ± 4.2	14.2 ± 7.3	17.0 ± 9.0	16.5 ± 9.8	12.9 ± 4.1	17.0 ± 8.4	16.9 ± 9.9	17.5 ± 9.8	17.5 ± 9.2	0.498	0.262	0.446
TW	L	34.8 ± 6.0	36.1 ± 6.2	35.7 ± 6.9	35.1 ± 6.6	36.0 ± 9.0	36.9 ± 7.2	38.0 ± 7.2	35.0 ± 5.4	34.8 ± 5.6	32.0 ± 3.9	32.0 ± 4.3	32.8 ± 3.1	0.240	0.326	0.029
BCM	kg	22.7 ± 5.5	23.6 ± 5.9	23.2 ± 5.9	22.0 ± 6.1	22.3 ± 5.7	21.7 ± 5.3	25.4 ± 8.7	22.7 ± 4.7	20.4 ± 3.2	20.6 ± 2.9	20.7 ± 2.7	20.9 ± 3.7	0.067	0.603	0.213
PhA	°	4.9 ± 1.0	4.9 ± 0.8	4.8 ± 0.6	4.6 ± 0.8	4.7 ± 0.8	4.4 ± 1.0	5.3 ± 2.2	4.8 ± 0.7	4.4 ± 1.0	4.7 ± 0.9	4.9 ± 0.8	4.8 ± 0.9	0.194	0.837	0.632
S. PhA	°	−0.9 ± 1.0	−0.9 ± 0.7	−1.0 ± 0.5	−1.0 ± 0.8	−1.3 ± 1.1	−1.5 ± 1.1	−1.7 ± 0.9	−1.4 ± 0.8	−1.1 ± 1.7	−1.0 ± 0.4	−1.0 ± 0.4	−0.7 ± 0.5	0.378	0.072	0.662

BMI, body weight index; WC, waist circumference; PhA, phase angle; FFM, fat-free mass; FM, fat mass; TW, total water; BCM, body cell mass; S. PhA, standardized phase angle. Values are expressed as mean ± standard deviation.

**Table 8 nutrients-16-01905-t008:** Quality of life depending on the assigned treatment.

		Standard Dose of DMB				High Dose of DMB				Placebo				*p*-Value		
		Baseline	1 Month	2 Months	3 Months	Baseline	1 Month	2 Months	3 Months	Baseline	1 Month	2 Months	3 Months	Time (t)	Treatment (T)	T × t
Global health status	Points	66.67 ± 12.42	66.67 ± 21.65	68.75 ± 13.91	69.79 ± 16.02	46.21 ± 15.97	41.67 ± 22.27	44.05 ± 19.07	40.28 ± 8.19	57.41 ± 23.36	73.81 ± 23.78	73.81 ± 16.27	78.57 ± 9.45	0.000	0.473	0.181
**Functional Scales**																
Physical	Points	94.67 ± 6.89	95.56 ± 6.67	95.83 ± 7.07	94.17 ± 7.51	90.3 ± 8.62	85.83 ± 12.57	89.52 ± 13.25	92.22 ± 6.55	97.04 ± 4.84	96.19 ± 7.56	99.05 ± 2.52	98.1 ± 5.04	0.083	0.117	0.228
Daily activities	Points	96.67 ± 7.03	98.15 ± 5.56	97.92 ± 5.89	97.92 ± 5.89	90.91 ± 11.46	89.58 ± 15.27	95.24 ± 8.13	91.67 ± 13.94	96.3 ± 7.35	92.86 ± 8.91	95.24 ± 8.13	100.00 ± 0.0	0.159	0.125	0.408
Emotional	Points	89.17 ± 14.19	89.81 ± 14.3	91.67 ± 11.79	89.58 ± 11.57	75.76 ± 19.88	70.83 ± 17.82	70.24 ± 20.33	61.11 ± 20.18	85.19 ± 15.47	91.67 ± 12.73	89.29 ± 16.47	82.14 ± 26.1	0.074	0.273	0.947
Cognitive	Points	93.33 ± 11.65	94.44 ± 11.79	97.92 ± 5.89	97.92 ± 5.89	80.3 ± 14.56	70.83 ± 21.36	73.81 ± 16.27	69.44 ± 12.55	85.19 ± 22.74	90.48 ± 13.11	90.48 ± 18.9	92.86 ± 13.11	0.010	0.586	0.427
Social	Points	75 ± 22.57	88.89 ± 16.67	87.5 ± 14.77	87.5 ± 17.25	60.61 ± 30.98	68.75 ± 41.25	71.43 ± 34.31	58.33 ± 31.18	81.48 ± 19.44	85.71 ± 15	92.86 ± 13.11	85.71 ± 20.25	0.108	0.018	0.936
**Symptomatic Scales**																
Fatigue	Points	24.44 ± 23.89	20.99 ± 23.2	12.5 ± 12.51	13.89 ± 21.21	42.42 ± 18.47	40.28 ± 18.72	44.44 ± 26.45	50 ± 26.06	38.27 ± 26.12	25.4 ± 19.99	23.81 ± 23.51	20.63 ± 28.28	0.092	0.044	0.307
Nausea and vomiting	Points	3.33 ± 7.03	3.7 ± 11.11	2.08 ± 5.89	4.17 ± 11.79	12.12 ± 18.4	10.42 ± 17.68	9.52 ± 13.11	5.56 ± 13.61	7.41 ± 12.11	2.38 ± 6.3	4.76 ± 8.13	0.0 ± 0.0	0.457	0.232	0.517
Pain	Points	16.67 ± 19.25	11.11 ± 18.63	8.33 ± 12.6	10.42 ± 12.4	24.24 ± 31.06	35.42 ± 43.13	38.1 ± 39.34	50 ± 39.44	14.81 ± 21.15	4.76 ± 12.6	7.14 ± 13.11	4.76 ± 12.6	0.176	0.678	0.304
Dyspnoea	Points	3.33 ± 10.54	3.7 ± 11.11	4.17 ± 11.78	0.0 ± 0.0	15.15 ± 17.41	12.5 ± 24.8	4.76 ± 12.6	16.67 ± 18.26	3.7 ± 11.11	4.76 ± 12.6	0.0 ± 0.0	9.52 ± 25.2	0.615	0.306	0.244
Insomnia	Points	26.66 ± 30.63	11.11 ± 16.67	16.67 ± 17.82	16.67 ± 25.2	24.24 ± 26.21	29.16 ± 33.03	14.28 ± 26.22	22.22 ± 27.21	25.92 ± 27.78	23.81 ± 25.2	28.57 ± 29.99	23.81 ± 31.7	0.757	0.787	0.731
Loss of appetite	Points	6.67 ± 14.05	7.41 ± 14.7	4.17 ± 11.78	4.17 ± 11.78	15.15 ± 22.92	12.5 ± 17.25	14.28 ± 26.22	22.22 ± 27.21	14.81 ± 24.21	38.09 ± 29.99	33.33 ± 33.33	23.81 ± 31.7	0.070	0.688	0.227
Constipation	Points	16.67 ± 23.57	3.7 ± 11.11	4.17 ± 11.78	4.17 ± 11.78	15.15 ± 27.34	0.0 ± 0.0	19.05 ± 26.22	16.67 ± 18.26	18.52 ± 29.39	9.52 ± 25.2	14.28 ± 17.82	9.52 ± 16.26	0.608	0.048	0.716
Diarrhea	Points	10.00 ± 16.10	7.41 ± 14.7	8.33 ± 15.43	8.33 ± 23.57	21.21 ± 26.97	29.16 ± 33.03	23.81 ± 31.7	16.67 ± 18.26	14.81 ± 24.21	23.81 ± 31.7	19.05 ± 26.22	23.81 ± 31.7	0.174	0.723	0.879
Financial difficulties	Points	6.67 ± 14.05	3.7 ± 11.11	4.17 ± 11.78	4.17 ± 11.78	15.15 ± 22.92	4.17 ± 11.78	19.05 ± 26.22	11.11 ± 27.21	3.7 ± 11.11	4.76 ± 12.60	4.76 ± 12.60	9.52 ± 16.26	0.821	0.195	0.192

**Table 9 nutrients-16-01905-t009:** Parameters of carbohydrate and lipid metabolism and nutritional status depending on the treatment.

		Standard Dose of DMB				High Dose of DMB				Placebo				*p*-Value		
		Baseline	1 Month	2 Months	3 Months	Baseline	1 Month	2 Months	3 Months	Baseline	1 Month	2 Months	3 Months	Time (t)	Treatment (T)	T × t
Glucose	mg/dL	106.1 ± 15.37	104.89 ± 12.72	100 ± 15.22	102.5 ± 13.9	108.82 ± 30.68	102.13 ± 15.81	95 ± 7.98	104.83 ± 26.01	109.3 ± 22.99	104.57 ± 27.96	106.86 ± 25.37	102.05 ± 23.21	0.809	0.678	0.789
Insulin	µU/mL	11.3 ± 10.71	10.56 ± 8.35	6.25 ± 3.33	6.21 ± 3.28	12.18 ± 7.67	9.14 ± 8.11	9.29 ± 6.65	16 ± 14.46	11.3 ± 9.02	9 ± 7.44	11.29 ± 8.98	10.1 ± 6.19	0.845	0.323	0.435
Total Cholesterol	mg/dL	187.4 ± 33.28	174.44 ± 38.76	170.63 ± 33.39	178.5 ± 27.15	174.45 ± 27.19	177.13 ± 27.22	186.24 ± 23.43	188.67 ± 36.46	185.2 ± 29.08	169.57 ± 26.51	183.29 ± 23.62	187.1 ± 21.69	0.751	0.932	0.768
HDL Cholesterol	mg/dL	60.7 ± 23.99	53.33 ± 23.08	50.88 ± 22.08	52.25 ± 25.42	53.45 ± 17.95	51.63 ± 17.72	50.19 ± 19.02	56.33 ± 21.71	57 ± 24.82	55.43 ± 24.58	64.14 ± 27.01	65.48 ± 28.25	0.909	0.546	0.192
No HDL	mg/dL	97 ± 0	121.11 ± 34.09	119.75 ± 31.67	131.83 ± 50.7	132.5 ± 0	125.5 ± 29.77	124.14 ± 41.33	132.5 ± 0	130.75 ± 27.93	114.14 ± 15.85	118.86 ± 27.27	126 ± 21.4	0.801	0.991	0.989
LDL Cholesterol	mg/dL	105.9 ± 24.12	99.67 ± 27.58	100.5 ± 26.44	106.13 ± 25.93	95.91 ± 30.45	99.38 ± 30.39	107.19 ± 22.49	111 ± 27.62	100.2 ± 25.19	91.29 ± 18.87	91.86 ± 19.18	97.86 ± 14.68	0.244	0.721	0.694
Triglycerides	mg/dL	113.8 ± 60.35	107.44 ± 49.51	96.38 ± 28.87	102.42 ± 43.73	145.45 ± 67.59	134.88 ± 66.2	145.38 ± 45.12	107.5 ± 26.36	140.2 ± 44.86	114.29 ± 30.51	135.57 ± 60.35	120.33 ± 50.46	0.203	0.58	0.506
Total proteins	g/dL	6.77 ± 0.45	6.68 ± 0.33	6.78 ± 0.31	6.53 ± 0.38	6.96 ± 0.7	6.79 ± 0.76	6.86 ± 0.57	6.95 ± 0.62	6.2 ± 1.47	6.66 ± 0.5	6.71 ± 0.34	6.79 ± 0.43	0.268	0.502	0.415
Albumin	g/dL	4.35 ± 0.17	4.22 ± 0.2	4.31 ± 0.2	4.06 ± 0.5	4.24 ± 0.31	4.1 ± 0.23	4.2 ± 0.21	4.32 ± 0.23	4.33 ± 0.36	4.17 ± 0.28	4.21 ± 0.23	4.27 ± 0.21	0.595	0.062	0.114
Prealbumin	mg/dL	23.8 ± 7.11	22.22 ± 5.47	20.47 ± 5.29	20.29 ± 5.76	22.07 ± 7.96	21.49 ± 8.34	23.73 ± 8.94	24.94 ± 11.42	19.94 ± 5.68	17.61 ± 3.76	19.74 ± 3.34	20.58 ± 4.55	0.337	0.297	0.152
RBP	mg/dL	4.36 ± 1.68	3.93 ± 1.47	3.79 ± 1.32	3.76 ± 1.64	4.22 ± 1.34	3.99 ± 1.54	4.29 ± 1.42	4.69 ± 2	3.88 ± 1.28	3.07 ± 0.6	3.33 ± 0.78	4.16 ± 1.32	0.47	0.218	0.027

RBP, retinol binding protein. Values are expressed as mean ± standard deviation.

**Table 10 nutrients-16-01905-t010:** Security parameters depending on the assigned treatment.

		Standard Dose of DMB				High Dose of DMB				Placebo				*p*-Value		
		Baseline	1 Month	2 Months	3 Months	Baseline	1 Month	2 Months	3 Months	Baseline	1 Month	2 Months	3 Months	Time (t)	Treatment (T)	T × t
Creatinine	mg/dL	0.76 ± 0.27	0.79 ± 0.27	0.85 ± 0.26	0.77 ± 0.28	0.71 ± 0.2	0.7 ± 0.18	0.68 ± 0.15	0.71 ± 0.25	1 ± 1.03	0.65 ± 0.11	0.68 ± 0.12	0.75 ± 0.18	0.329	0.033	0.054
GFR	mL/min/1.73 m^2^	82 ± 13.22	79.44 ± 13.74	78.38 ± 15.83	81.17 ± 14.03	87.55 ± 8.68	89.63 ± 3.89	90.14 ± 2.27	87.17 ± 9.39	81.1 ± 23.74	88.14 ± 7.56	86.57 ± 8.42	81.71 ± 14.19	0.832	0.202	0.051
Uric acid	mg/dL	4.25 ± 1.27	4.56 ± 1.77	4.86 ± 2.08	4.12 ± 1.52	4.38 ± 1.2	4.59 ± 0.82	4.64 ± 1	4.55 ± 1.22	5.1 ± 2	4.34 ± 1.13	4.77 ± 0.98	5.01 ± 1.28	0.16	0.057	0.066
AST	UI/L	31.64 ± 12.61	29.5 ± 18.73	27.1 ± 18.42	26.39 ± 12.27	29.37 ± 17.57	30.96 ± 22.07	35.86 ± 30.03	31.67 ± 21.56	26.75 ± 15.23	26 ± 10.77	28.29 ± 20.31	28.05 ± 14.15	0.778	0.985	0.903
ALT	UI/L	34.2 ± 18.02	24.33 ± 12.86	27.88 ± 15.01	23.88 ± 12.8	26.64 ± 13.31	25.13 ± 18.91	31.19 ± 24.55	29.5 ± 13.77	32.11 ± 11.6	32.71 ± 14.51	36.43 ± 16.52	37.71 ± 17.98	0.221	0.051	0.057
LDH	UI/L	225.6 ± 25.68	230.75 ± 43.44	238.75 ± 27.86	229.25 ± 23.75	291.8 ± 154.17	236.08 ± 77.87	272.07 ± 109.24	240.08 ± 117.92	520.74 ± 841.91	258 ± 65.68	262 ± 53.59	248.11 ± 60.64	0.267	0.865	0.996
AP	UI/L	82.57 ± 27.28	79.17 ± 20.15	69.8 ± 9.65	72.07 ± 10.39	149.6 ± 138.72	145.56 ± 172.63	167.79 ± 199.69	89 ± 31.03	146.78 ± 75.96	128.14 ± 43.13	131.14 ± 46.46	126.48 ± 40.81	0.264	0.934	0.982
GGT	UI/L	39.14 ± 20.31	35.33 ± 19.19	31 ± 17.07	34.07 ± 18.74	168.3 ± 389.02	235 ± 523.74	250 ± 544.6	43.17 ± 29.39	142.63 ± 157.81	92.33 ± 92.83	117.67 ± 129.16	116.06 ± 122.14	0.537	0.727	0.748
Bilirubin	mg/dL	0.74 ± 0.45	0.66 ± 0.37	0.6 ± 0.24	0.65 ± 0.36	0.53 ± 0.13	0.52 ± 0.3	0.47 ± 0.24	0.46 ± 0.15	0.56 ± 0.31	0.43 ± 0.2	0.52 ± 0.19	0.5 ± 0.22	0.239	0.730	0.867

GFR, glomerular filtration rate; AST, aspartate aminotransferase; ALT, alanine transaminase; LDH, lactate dehydrogenase; AP, alkaline phosphatase; GGT, gamma-glutamyl transferase. Values are expressed as mean ± standard deviation.

## Data Availability

The data used to support the findings of this study are available from the corresponding author upon request.

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
