# Peer review of "Efficacy and Safety of Habitual Consumption of a Food Supplement Containing Miraculin in Malnourished Cancer Patients: The CLINMIR Pilot Study"

_nutrients, 2024, doi:10.3390/nu16121905_

Round 1
Reviewer 1 Report
Comments and Suggestions for Authors
Good job. Very interesting article:) I hope that in the future it will be possible to conduct it on more patients:) I have a few minor comments.
line 42-43: so they consumed DMB/placebo once a day?
line 78-79: "One of the most prevalent TDs is dysgeusia, which occurs between 56% and 76% of patients receiving antineoplastic treatment [14]." TDs and dysgeusia are not the same thing?
line 171: "one holiday" I don't understand
line 349: "10 cm scales". cm? maybe it was a 10-point scale...
line 619-620: "was normal just in those patients consuming the standard dose of DMB while the rest were above normal ranges (< 38 IU/L)." shouldn't it be (> 38 IU/L)?
Author Response
We are especially grateful for your comments and suggestions, which have undoubtedly contributed to significantly improving our manuscript.
- Line 42-43: so they consumed DMB/placebo once a day?
Thanks again for your thoughtful comments. Over 3 months, each patient consumed an orodispersible tablet containing DMB or placebo five minutes before each main meal (breakfast, lunch, and dinner). This information appears in manuscript section 2.3 Interventions. However, for more clarity, we have completed the abstract indicating the number of meals per day in which DMB was consumed as follows:
Line 44: Patients daily consumed a DMB tablet or placebo before each main meal (breakfast, lunch, and dinner).
- Line 78-79: "One of the most prevalent TDs is dysgeusia, which occurs between 56% and 76% of patients receiving antineoplastic treatment [14]." TDs and dysgeusia are not the same thing?
From the qualitative point of view, there are different concepts. Taste disorders (TD) are a condition characterized by an alteration in gustatory function or perception. These include dysgeusia but also hypogeusia (decreased sensitivity to taste modalities), hypergeusia (increased sensitivity), phantogeusia (phantom taste i.e., taste can be perceived without an external stimulus) and ageusia (loss of taste). The most common taste disorder qualitatively speaking by far is dysgeusia. Dysgeusia, a taste disorder typically characterized by an unpleasant and persistent taste, is often described as metallic, and other taste disorders, which are common among cancer patients. To a better understanding we have added some information as follows:
Line 82: One of the most prevalent TDs, from the qualitative point of view, is dysgeusia, which occurs between 56% and 76% of patients receiving antineoplastic treatment [14]. Dysgeusia is a qualitative gustatory disturbance defined as impaired or altered sense taste perception or persistent taste sensation without stimulation [15].
- line 171: "one holiday" I don't understand
It refers to a holiday as a weekend day, a day off, or a day out of the usual routine. For greater understanding, this clarification has been added in section 2.3 Interventions (Line 171).
- Line 349: "10 cm scales". cm? maybe it was a 10-point scale...
It is a metric scale. The visual analog scale (VAS) is usually a 10 cm long metric scale to accurately measure the score given by the individual using a ruler (Figure). Effectively, when the measurement is finished can be expressed in points but the scale has a length of 10 cm.
- Line 619-620: "was normal just in those patients consuming the standard dose of DMB while the rest were above normal ranges (< 38 IU/L)." shouldn't it be (> 38 IU/L)?
Dear reviewer, you are right, thank you for your observation. We have changed the symbol.

Reviewer 2 Report
Comments and Suggestions for Authors
An exploratory clinical trial was carried out in which 31 cancer patients were randomized into three arms [standard dose of DMB (150 mg DMB/tablet), high dose of DMB (300 mg DMB/tablet) or placebo (300 mg freeze-dried strawberry)] for three months. The electrochemical taste perception, nutritional status, dietary intake, quality of life and the fatty acid profile of erythrocytes were evaluated. However, there are some weaknesses that need to be addressed.
1. Line 38: DMB means a novel supplement containing the natural taste-modifying protein miraculin, Line 112: DMB represents dried miracle berry. It is confusing what is the DMB?
2. Figure 2: the calculated participants in the last step “Analysis” in the three groups were wrong. Only 21 cancer patients completed the clinical trial (Line 377-378).
3. Line 155-158: The first arm had 150 mg of DMB equivalent to 2.8 mg of miraculin +150 mg of freeze-dried strawberries per orodispersible tablet; the second arm had 300 mg of DMB equivalent to 5.6 mg of miraculin; and the third arm contained 300 mg of freeze-dried strawberries per orodispersible tablet as a placebo.
Why do you choose 150 mg DMB as the standard dose group and 300 mg as high dose group? Provide the basis of the two doses.
Why do you choose 300 mg of freeze-dried strawberries as placebo, not 150 mg of freeze-dried strawberries? In the standard dose group, why do you add 150 mg freeze-dried strawberries? As shown in Table 1, the nutritional composition and content in the three groups were different and there are no multiple relationships in nutritional composition except Miraculin. Therefore, how to prove that the improved taste acuity or salty taste perception was related only a standard dose of DMB, not the combination of DMB and freeze-dried strawberries?
4. Figure 4: please provide the standard deviation in the figure.
5. Line 460-462: a p-value < 0.05 was considered statistically significant (Line 366-367). Therefore, there was no significant change over time and depending on treatment in monounsaturated fatty acids (MUFA, p = 0.092).
6. Line 483: what is the meaning of “AA”?
7. Line 584: provide Table S5 in the supplementary.
Author Response
We thank very much the reviewer #2 for his/her thoughtful comments, which have contributed to improving our manuscript significantly
- Line 38: DMB means a novel supplement containing the natural taste-modifying protein miraculin, Line 112: DMB represents dried miracle berry. It is confusing what is the DMB?
Dear reviewer, thank you for your observation. DMB is the acronym for dried miracle berries, a novel food approved by the European Commission for consumption in the European Union. DMB is also available as a supplement. For better understanding, we have modified the manuscript in the abstract section as follows:
Line 37: The novel food approved by the European Commission, dried miracle berries (DMB), contains the natural taste-modifying protein miraculin. DMB, also available as a supplement, has emerged as a possible alternative treatment for TDs.
In the manuscript introduction section (Line 112) is the extended explanation of what DMB is.
- Figure 2: the calculated participants in the last step “Analysis” in the three groups were wrong. Only 21 cancer patients completed the clinical trial (Line 377-378).
Effectively, only 21 patients have completed the clinical trial. For this reason (reduced sample size), an intention-to-treat analysis was performed and not a per-protocol analysis. Intention to treat analysis is the gold standard for randomized clinical trials. In this analysis, data from all subjects initially enrolled in the clinical trial are used for the analysis of efficacy and safety regardless of whether they completed or received the treatment, to preserve randomization.
The analysis method used in the study is indicated in 2.13 Statistical methods (Line 362) and again noted in the 3. Results section (Line 382).
- Line 155-158: The first arm had 150 mg of DMB equivalent to 2.8 mg of miraculin +150 mg of freeze-dried strawberries per orodispersible tablet; the second arm had 300 mg of DMB equivalent to 5.6 mg of miraculin; and the third arm contained 300 mg of freeze-dried strawberries per orodispersible tablet as a placebo.
Why do you choose 300 mg of freeze-dried strawberries as placebo, not 150 mg of freeze-dried strawberries?
This was because we had a double dose of 300 mg of DMB. To mask the appearance of tablets (pinky color) of the three products, all had to have the same weight (300 mg) although the proportions between DMB and freeze-dried strawberries were different. In both DMB and freeze-dried strawberries, color is mostly provided by anthocyanins. Also, we used dried strawberries to match the polyphenols content and the taste of the three intervention tablets as DMB has about the same acidity as freeze-dried strawberries.
In the standard dose group, why do you add 150 mg freeze-dried strawberries? As shown in Table 1, the nutritional composition and content in the three groups were different and there are no multiple relationships in nutritional composition except Miraculin.
In addition to the weight, the freeze-dried strawberries provided acidity, color and aroma to the standard tablet. This made the tablets indistinguishable from each other, creating a perfect masking of the study products (Figure). Additionally, the incorporation of freeze-dried strawberries to the standard tablet allowed the products to be isocaloric.
Therefore, how to prove that the improved taste acuity or salty taste perception was related only a standard dose of DMB, not the combination of DMB and freeze-dried strawberries?
We attribute these changes to DMB because of the freeze-dried strawberries cannot modify the taste and DMB, which contains miraculin, does. In general terms, freeze-dried strawberries could provide some antioxidants but do not affect changes in taste perception.
- Figure 4: please provide the standard deviation in the figure.
Thank you very much for your suggestion, the SDs have been incorporated into Figure 4.
- Line 460-462: a p-value < 0.05 was considered statistically significant (Line 366-367). Therefore, there was no significant change over time and depending on treatment in monounsaturated fatty acids (MUFA, p = 0.092).
Dear reviewer thank you for your observation. The sentence has been changed as follows:
Line 466: Moreover, there was a significant change over time and depending on treatment in the dietary percentage provided by saturated fatty acids (SFA, p = 0.042) and a trend in monounsaturated fatty acids (MUFA, p = 0.092).
- Line 483: what is the meaning of “AA”?
AA is Arachidonic Acid. It has been added the meaning in the manuscript as follows:
Line 487: In the groups consuming standard and high doses of DMB, arachidonic acid (AA) increased the percentage change by 49.9 ± 57.9 and 42.1 ± 49.8 %, respectively, while in placebo 8.4 ± 31.5 %.
- Line 584: provide Table S5 in the supplementary.
Thank you very much for your observation, we had not realized we had missed it. It has been integrated into the supplementary material.
